# Development of Stainless Steel Yarn with Embedded Surface Mounted Light Emitting Diodes

**DOI:** 10.3390/ma15082892

**Published:** 2022-04-14

**Authors:** Abdella Ahmmed Simegnaw, Benny Malengier, Melkie Getnet Tadesse, Lieva Van Langenhove

**Affiliations:** 1Department of Materials, Textiles and Chemical Engineering, Ghent University, 9000 Gent, Belgium; benny.malengier@ugent.be (B.M.); lieva.vanlangenhove@ugent.be (L.V.L.); 2Ethiopian Institute of Textile and Fashion Technology, Bahir Dar University, Bahir Dar 1037, Ethiopia; melkiegetnet23@gmail.com

**Keywords:** smart textile, wearable textile, conductive yarn, E-yarn, surface mounted devices, light emitting diode, e-textile

## Abstract

The integration of electronic components in/onto conductive textile yarns without compromising textile qualities such as flexibility, conformability, heat and moisture transfer, and wash resistance is essential to ensuring acceptance of electronic textiles. One solution is creating flexible and stretchable conductive yarns that contain tiny surface-mounted electronic elements embedded at the fiber level. The purpose of this work was to manufacture and subsequently evaluate the physical features and electromechanical properties of stainless steel yarn with light-emitting surface mounted devices (SMDs) embedded in it. The SMDs were successfully integrated into a conductive stainless steel yarn (SS) by inserting crimp beads and creating a bond through hot air soldering machines, resulting in what we call an E-yarn. The relationship curves between gauge length and electrical resistance, and the relationship curves between conductive yarn elongation and electrical resistance, were explored experimentally. The results of the analysis demonstrated that E-yarn had a lower tensile strength than the original electrically-conductive SS yarn. The effects of the washing cycle on the conductivity of the E-yarn were also investigated and studied. The results showed that E-yarns encapsulated at the solder pad by heat shrink tube still functioned well after ten machine wash cycles, after which they degraded greatly.

## 1. Introduction

### 1.1. Smart Textile

Smart textiles are a new generation of textile that play critical roles in a wide range of technical applications. Smart and intelligent textiles refer to a new generation of fibers, fabrics, and materials that can detect and react to changes in the environment, such as mechanical, chemical, thermal, magnetic, electrical, and optical changes, in a predetermined way [1,2,3]. For practical appliances, smart textiles have five purposes. They can operate as sensors, data processing elements, actuators, power storage devices, and communication devices [4]. 

Smart textiles can be constructed in numerous forms. Most commonly, smart textiles have been created in the form of electronic textiles (e-textiles) [5]. E-textiles are textile yarns or textile fabrics in which electronics and circuits are implanted. The oldest technique for embedding conductive materials into a textile substrate to form conductive fiber networks or circuits is to incorporate tiny conductive metals, e.g., by twisting them into the textile yarns [6]. However, thin metals cannot withstand abrasion, especially during laundry cycles, so more solutions that are robust are required. 

Integration of electrical components such as LEDs, resistors, thermistors, capacitors, and inductors on or in conductive fabrics is a major application area of wearable e-textiles and has been extensively researched in recent decades [7]. For the integration, it is necessary to construct reliable textile-based transmission tracks during the first phase of the e-textile structure construction. Researchers have tried several mechanisms of interconnection of electronic device with textile substrates, as discussed in the following section.

### 1.2. Methods of Integration

Several researchers have conducted studies on joining methods for electrical components and textile fabrics using various techniques, by highlighting their merits and drawbacks. These techniques included mechanical interconnection, stitching and sewing, embroidering, soldering and adhesives [8,9].

#### 1.2.1. Mechanical Interconnection

Various methods of mechanical interconnection of electronics and textiles have been used, such as snapper plugs, zippers, hook and loop connectors, and magnetic connectors [10]. These either use rigid parts or are not suitable for many connections due to their bulk or weight, thereby becoming burdensome. Furthermore, due to the reduced dimensional rigidity of the components, the freedom of motion by a fabric substrate is restricted [11].

#### 1.2.2. Sewing and Embroidery

Functional smart textiles have been developed by embedding SMDs using a Brother PR-650e embroidery machine [12]. The results showed that failure occurred at the connection point between the stitched thread and the SMD, which resulted in poor performance of the product. Researchers have effectively integrated SMD onto a textile fabric in the form of bendable electronic sheets in sequin [13]. In addition, stitching has been used to attach SMDs onto a fabric substrate [14]. However, the uninsulated conductive thread was susceptible to corrosion during washing, and subsequently the added plastic insulation base altered the material’s feel. Furthermore, the LEDs were prone to breaking off during twisting. In addition to stitching, subtractive technologies and 3D printing have been used to make flexible and functional sequins for handheld embroidery textile applications [15]. Electronic sequins can be embroidered or stitched onto textile fabrics with electrically conductive thread. A conductive thread, however, is susceptible to mechanical abrasion. 

#### 1.2.3. Adhesives

Electronic components have been attached onto a textile circuit with a non-conductive glue using pressure force to transfer the glues and make a secure mechanical bond and electrical connection. The thermoplastic adhesive adhered the electronics to the fabric by applying pressure and heat [16]. Li and Wong [17] worked with conductive glues to replace solder in electronic packaging applications. However, this type of technology is not widely available and has a number of drawbacks, including poorer electrical conductivity, little conductivity fatigue resistance, and a limited lifespan. 

##### 3D Printing

3D printing has also been used to make electrical and mechanical contacts between SMD electronic parts and textiles [18]. However, the contact resistance between 3D-printed conductive tracks and conductive yarns was too high for practical usage and should be lowered. Furthermore, the contact between the 3D printed area and the electronic components that were afterwards inserted was still not satisfactory.

#### 1.2.4. Soldering

Buechley and Eisenberg [19] studied the technique of integrating fabric and PCBs via soldering using silver crimping bead attachments onto the terminals of surface-mounted LEDs with lead-free solder. However, needs further processing steps are needed for the LED sequins before they can be sewn onto the textile fabric with conductive thread. The crimp beads can be soldered to the SMD perpendicularly, allowing a thread to move through it and be crimped. In addition, the interconnection of electronic components with a smart textile has been done using pulsed laser soldering techniques [20]. The results showed that the textile-integrated, varnish-insulated copper strings had successful contacts. However, the nearby polyester knitted fabric was heated and melted. From this, it can be seen that the interconnection of electronic devices and textile substrates, especially textile fabrics, can be successfully performed. However, the integration still suffers from inconsistencies between smooth, elastic, pliable textile fabrics and stiff electronic components, particularly because sequins are already larger than the pure SMD. This has a significant impact on the final design and properties of the textile fabric. 

In order to accomplish higher degrees of incorporation and user comfort, scholars have developed and studied stretchable and bendable electronic parts for smart textile applications [21]. In addition, the development of E-yarns by integrating miniaturized electronics into textile fibers or yarns has also been carried out. E-yarn with integrated electrical components can be used as a building block for smart textiles, allowing for the creation of electronically functional wearable textiles, for instance, wearable sensors, wearable heaters, and wearable color changing displays [22]. The difficulty with these is the correct positioning of the E-yarn so that the component is in the correct position, as the component is present in the yarn before the yarn is placed in the circuit, as opposed to the methods presented which form the circuit first and then attach a component at each necessary position.

### 1.3. Integration of Electronics at Yarn Level

A recent development by Rein et al. [23] developed LEDs integrated into copper metallic wires or high-melting-temperature tungsten during the extrusion process of drawing a fiber. Due to the impossibility of extracting the filaments quickly after extrusion without destroying the copper wire connections, the extrusion technique produces a yarn with low tensile strength [24]. This has an impact on the copper wire’s capacity to be processed in normal textile manufacturing procedures. Moreover, it is limited to conductive materials for co-drawing, and it is also limited to electronic devices with two terminals. This will limit the number of electronic devices that can be used and the functionality of E-yarn.

A method for embedding SMDs on 2 mm wide flexible plastic strips has been devised by the ETH Zurich wearable computing laboratory [25,26]. In the weft direction, the components were weaved into a cloth. However, the degree of bending of the strips was limited due to the use of a typical bare die, and the strips were not ideal for knitting or embroidery. Combinations of electronics on stretchy plastic strips with textiles, during the weaving process, have also been investigated. 

The E-thread^®^ [27] was developed by the PASTA project, in which the die was embedded into two conductive wires and the die and interconnects were masked with a fibrous blanket. This E-thread^®^ was not appropriate for washing, and the chips were not enclosed. Only two-pin electronic devices could be used in the E-Thread^®^ manufacturing, and this limited the range of electronic devices and thus the functions that could be incorporated.

An E-yarn was developed consisting of a conductive core of multi-stranded copper wire to which semiconductor dies were attached by direct soldering [28]. The E-yarn was then covered in a mastic micro-pod, which was successively surrounded by a textile casing that also protected the copper wires. Due to their copper conductors, they either were not bent at a 90° angle or had a limited number of bending cycles [29]. In turn, the thickness of the sealing material led to an increase in the diameter of the E-yarn. Large yarn diameters reduce the comfort of the textile. They can also hinder moisture and heat transfer, which are critical performance properties [30].

The development and electro-mechanical characterization of e-conductive yarns are important for their subsequent applications in wearable e-textiles [31,32]. The electrical resistance of a conductive yarn is a critical design factor for E-yarns [33]. Besides investigating their electro-conductive properties, there is a need to investigate their electro-mechanical behavior for different applications of smart textiles [34,35].

In e-textile, low linear resistance E-yarns are used to transfer electrical signals. However, these yarns are subjected to significant strains during production or usage, which can lead to conductive track failure. Therefore, for the design and fabrication of safe and reliable e-textiles, electrically conductive textiles made from conductive yarns require thorough characterizations of their properties. For this reason, the physical and electromechanical behavior of the E-yarn should be investigated. 

The physical and mechanical properties (tensile strength and elongation) and electrical properties of conductive textile yarns were studied in [36,37]. Furthermore, the electromechanical properties of textile structures for wearable sensors made of silver-coated conductive Statex^®^ yarn were also studied [38,39].

Overall, the research done so far is limited, and therefore, there are inadequate data on the electrical and electromechanical analyses of E-yarns, especially flexible, conductive yarns with SMD light emitting diode (LED) electronics embedded, which are more complex components to characterize than resistors.

The purpose of this study was to develop a functional SMD LED-embedded E-yarn using commercially available stainless steel conductive yarn and miniature SMDs. Hot air soldering methods were used to establish the integration between each micro-SMD and the stainless steel yarn. Afterward, the features, physical and electromechanical properties of the E-yarn were investigated. The resulting E-yarn can be used for fashion, safety equipment or light treatment garments.

## 2. Materials and Methods

### 2.1. Materials

In this experiment, a commercially available BEKINOX, VN.12.2.2.175 stainless steel (SS) multi filament, 2/1 ply, 555 Tex conductive yarn manufactured by the bundle drawing process was used. These conductive threads were purchased from BEKAERT (Belgium) and were halogen free, flexible, durable, and corrosion resistant. They can be easily woven, knitted, sewn, or embroidered. In addition, white SMD 8000 K (Assembly chip DC3 3.4 V, 150 mA) super bright LEDs (light emitting diodes) with dimensions 1.6 × 1.2 × 2 mm (L × W × D) were used. The selection of the SMD type was based on the stainless steel conductive yarn’s properties, such as fineness, flexibility, low weight, and relatively low cost. For assembling purposes, 1.5 mm cylindrical silver crimp beads were used. Furthermore, a 2 mm heat shrink tube served to insulate the connection point of the SMD and the stainless steel yarn, as shown in Figure 1.

### 2.2. Methods

#### Development Process of Stainless Steel Yarn with Embedded SMD Components

The SMD electronic components were selected to be integrated into stainless steel conductive thread. For connecting the SMD with a SS conductive thread, a soldering method was applied. However, direct soldering of stainless steel yarn is difficult due to the existence of a thick passive oxide (Cr_2_O_3_) film that blocks the melted solder paste from sticking to the surface of the SS textile thread [40]. Therefore, other additional alternative techniques of direct soldering were required. 

A unique integration method was developed. First, a surface preparation of the stainless-steel yarn was performed. The tips of the SS conductive yarn were heated via 50 °C hot air for 2 min and a small drop of 85% concentrated H_3_PO_4_ paste (phosphoric acid-based paste) was applied. The surface preparation was used to polish the thick oxide layer at the surface of SS thread. In addition, it helps with activating the SS thread in order to connect and stick them together for the next step. A proper connection of the SMD and conductive SS yarn with the 1.5 mm silver crimp beads was performed by applying a mechanical force by using pliers on the bead after inserting the SS tip. Furthermore, the final integration of the SMD into the stainless steel conductive thread was done by using a four-in-one 909 D hot air gun soldering rework station, immediately after cleaning. Thereafter, to create a protective layer on the solder pad, a 1.5 mm heat shrink tube was transplanted over at the connection joints. The steps used to create the E-yarn are shown in Figure 2.

The soldering steps in Figure 2: Two pieces of 15 cm stainless steel yarn to be connected at both ends into crimp beads were cut.The tip of the stainless steel thread was bent to form a loop, which was inserted into the 1.5 mm cylindrical crimp beads. The loop at the tip improved the connection in the crimp bead.The crimp bead was pressed securely by using flat nose pliers, to flatten and hold the yarn firmly.The terminals of the SMDs were connected to the tip of flat crimp beads by adhesive tape.Solder paste was applied on the connection point of the SMD and crimp beads.Controlled temperature was applied with hot air onto the soldering paste, which melted without damaging the component, thereby connecting the crimp beads to the pads of the SMD.As an optional extra step, a 2 mm heat shrink tube was placed over the joints, and hot air was applied to shrink them and create an insulating layer over the solder joints.

### 2.3. Experimental Setup

#### 2.3.1. Measurement of the Linear Electrical Resistance of the SS Conductive Yarn

The electrical properties of SS conductive yarn samples were determined by measurement of current–voltage curves. The length-dependent electrical resistivity of a sample was measured with a four-point conductivity measurement probe and evaluated at variable lengths of 0.05, 0.1, 0.15, 0.2, 0.25, 0.3, 0.35, 0.4, 0.45, and 0.5 m with 10 repeats. For each specimen, the average was taken for analysis. A standardized method for conductivity measurements was performed according to the AATCC TM84 [34] test standard by using both digital multimeter and Burster clamps four-point resistance measuring, as shown in Figure 3.

#### 2.3.2. Measurement of the Stress-Dependent Electrical Resistance of SS Conductive Yarn

Investigating the electrical resistance of the conductive textile thread under load is an essential factor for the applications where the smart textile is exposed to stress and bending. The electrical resistance associated with extension under tensile stress at room temperature was examined using the four-point probe technique. The stress-dependent electrical resistance of the stainless steel yarn and SMD embedded E-yarn was investigated using a Mesdan universal strength tester machine under the guidelines of ISO, ISO 2062:2009 [41], with 200 mm gauge length. A pretension 2N force was applied with a loading speed of 20 mm/min. The conductive yarn was strained in its longitudinal direction, and the measurement of the stress-based electrical resistance of the conductive thread was done by attaching the electrodes of the multimeter at both terminals of the SS conductive yarn specimen. Five samples were measured, and the average resistance was recorded. 

#### 2.3.3. Measurements of Total Electrical Resistance of E-Yarn 

An LED circuit, commonly called an LED driver, is an electronic circuit that supplies power to a light emitting diode (LED). This circuit must carry adequate current to turn on the LED to the appropriate brightness while also preventing the current from damaging the LED. With burster clamps, four-probe conductivity measurements were used to determine the overall electrical resistance of the E-yarn. The resistance was tested on a 30 cm stretch of E-yarn with one SMD implanted every 15 cm. The current and voltage flows in LED circuits do not have a linear relationship. They cannot be modeled as governed by Kirchhoff’s second law of the association between the voltage drop across the loop. However, the resistance of the LED can be expressed using a piecewise linear model [42]. Figure 4 depicts the voltage drop across each node in the circuit, and the voltage drop can be determined using Equation (1). Furthermore, the overall resistance of the SMD embedded E-yarn was a function of the conductive yarn resistance, solder connector resistances, and SMD resistor resistances.
(1)$$V_{S}=V_{\mathrm{SSL}}+V_{\mathrm{LC}}+V_{\mathrm{LED}}+V_{\mathrm{RC}}+V_{\mathrm{SSR}}\;\mathrm{or}\;V_{S}={\mathrm{IR}}_{\mathrm{SSL}}+{\mathrm{IR}}_{\mathrm{LC}}+V_{\mathrm{LED}}+{\mathrm{IR}}_{\mathrm{RC}}++{\mathrm{IR}}_{\mathrm{SSR}},$$
where $V_{\mathrm{source}}$, denotes voltage source; $V_{\mathrm{SSL}}$ and $V_{\mathrm{SSR}}$ is the voltage drop in left and right conductive yarns, respectively; $V_{\mathrm{LC}}$ and $V_{\mathrm{RC}}$ are the voltage drop at left and right connectors; and $V_{\mathrm{LED}}$ represents the forward voltage drope at the SMD LED. Furthermore, I is the current flow in the circuit, $R_{\mathrm{SSL}}$ denotes the resistance of the left conductive yarn. $R_{\mathrm{LC}}$ is the resistance of left solder pad connector, $R_{\mathrm{LED}}$, is the approximate resistance of the LED at a specific input voltage and forward current, and $R_{\mathrm{SSR}}$ is the electrical resistance of the right conductive yarn.

The LED characteristic curve was approximated as a series of linear segments, using a range of current flows in the circuit set with a voltage source. This enabled us to model the LED as a resistor with a forward voltage (V) and forward current (I), and utilize Ohm’s law to obtain an LED as resistor, as shown in Figure 4. We have: (2)$$R_{\mathrm{LED}}\left(I_{\mathrm{LED}}\right)=\;\frac{V_{\mathrm{LED}}}{I_{\mathrm{LED}}},$$
where $V_{\mathrm{LED}}$ denotes the forward voltage drop at SMD LED and $I_{\mathrm{LED}}$ represents the current in the circuit corresponding to this voltage drop. 

As a result, the total resistance of the SMD integrated E-yarn is a function of the current and depends further on the resistance of the conductive yarn and resistance of the connectors. The total resistance of the SMD LED-embedded E-yarn was computed via Equation (3).
(3)$$R_{T}=R_{\mathrm{SSL}}+R_{\mathrm{LC}}+R_{\mathrm{LED}}+R_{\mathrm{RC}}+R_{\mathrm{SSR}}\;\mathrm{and}\;R_{\mathrm{SSL}}+R_{\mathrm{LC}}+R_{\mathrm{RC}}+R_{\mathrm{SSR}}=\frac{\left(V_{\mathrm{source}}-V_{\mathrm{LED}}\right)}{I_{\mathrm{LED}}},$$
where this last expression is a constant, independent of the current or applied source voltage $V_{\mathrm{source}}$.

#### 2.3.4. Measurements of the Power of SS Conductive Yarn and E-Yarn

In addition to electrical resistance, power is a primary purpose of transmission lines in wearable electronics. Since the SS conductive yarn acted as a resistor in the circuit, as shown in Figure 4, the power dissipated at the conductive threads and also at the SMD LED are important factors in Joule heating. The output heat power (W) produced by a conductor material is proportional to the product of its resistance R (Ω) and the square of its direct current I, according to Lenz–Joule law. The experiment was verified by applying a variable voltage to the two ends of the SS conductive thread and the E-yarn using a four-point probe clamping device with an EL301R power supply source. The voltage drop between the two ends was measured by a 0.001 V-accuracy multimeter. Afterwards, the resistance at each node was calculated by using Equation (1). Five samples were examined with a variable input voltage (V_source_), from 0 V to the registered failure of current flow in the circuit. For the power, we have
(4)P=I V, 
where P is power, I is the current flow in the circuit, and V is the voltage. The power in each part of the circuit can be obtained. For the yarn and the contact points, this will be dissipated as heat; for the LED it will be dissipated as heat and emitted light, with an efficiency of 30% to 70%, depending of the type of LED and current applied. 

#### 2.3.5. Effect of Washing

The samples were washed for up to 25 washing cycles in a domestic washing machine with standard BS EN ISO 6330:2012 [43], to assess the usability and performance of the SMD embedded E-yarn. The machine was packed with cotton polyester blended textiles in addition to the conductive yarn and E-yarn samples to attain its standard weight of 2 kg. After each wash cycle, the samples were dried at room temperature for 12 hr For each cycle, three sample readings were taken, and the electrical resistance was determined by applying 3.0 V using the four-point probe. The change in resistance was computed using a resistance ratio to determine a trend and the impacts of laundering on the resistance properties of both the SS conductive yarn and the SMD embedded E-yarn: (5)$$\mathrm{CR}\%=\frac{{\;R}_{f}-R_{i}}{R_{i}}\times 100$$
where R_i_ is the measured resistance of the SS conductive yarn or E-yarn before washing, and R_f_ is the measured resistance after washing.

#### 2.3.6. Measurement of Tensile Properties

The tensile strength of stainless steel conductive yarn and fabricated SMD integrated E-yarns was measured using the Mesdan universal yarn strength tester. The tensile test was carried out following the procedures of ISO: ISO 2062:2009 [44]. According to the standard, the sample breaking time was achieved at the extension rate of 20 mm/min at the constant rate of an elongation type machine. The maximum loading cell capacity used was 500 N, along with a gauge length of 200 mm. Tensile tests were implemented on five samples of SS conductive yarns and five samples of E-yarns. Finally, the comparison of tensile characteristics, i.e., tensile strength, elongation, tenacity, and initial modulus, of the conductive yarn and SMD embedded E-yarn was investigated and analyzed.

## 3. Results and Discussion

### 3.1. Electrical Conductivity

Regarding the electrical properties, the conductive yarn’s electrical resistance was measured, and the resistance of the conductive yarn was derived from it, based on the yarn’s length. Figure 5 depicts the relationship between yarn resistance and yarn gauge length for individual samples at gauge lengths of 0.05, 0.1, 0.15, 0.2, 0.25, 0.3, 0.35, 0.4, 0.45, and 0.5 m measured using the four-point probe method. The nearly linear trend is clearly visible, indicating that the stainless steel conductive thread had a uniform makeup with nearly constant resistance per unit length.

The gauge length has a direct impact on the resistance of conductive yarn [44]; that is, the resistance is length-dependent. In Table 1 and Table 2, the experimental results show that the dependence can be approximated with a linear function, R_SS_*(x)* = 14.41 *x* − 0.028, with *x* being the gauge length. The slope is the most important parameter for characterizing the conductivity of the yarn. The p-value for the slope is < 0.001 (statistically significant), and the p-value of the intercept is 0.989 (not significant). In this case, the correlation coefficient of the curve is R^2^ = 0.99.

### 3.2. Stress-Dependent Electrical Resistivity of Stainless Steel Yarn

The measured electrical resistance and the quadratic fitting curve of the SS conductive yarn under tensile force are shown in Figure 6.

The effect of applying strain (relative elongation) on the electrical resistance of SS conductive yarn is demonstrated in Figure 6. The electrical conductivity decreased with increasing extension [45,46,47,48,49]. The results show that the electrical resistance of a 0.2 m gauge length piece of SS was a quadratic increasing function of elongation, given by R_SS_(*x*) = **−**0.021*x*^2^ + 0.195*x* + 2.85, with *x* being the extension, and with *p*-values of the coefficients <0.001, as shown in Table 3 and Table 4. It is clear that the electrical resistance of samples increased when the cross-sectional yarn area decreased or the extended length increased [32]. One would expect that the more aligned the structural elements (microfibrils) are, the larger the free path would be for electron movement, and that different fibers would have less contact resistances as they more firmly pressed upon one another, leading to lower resistance, but this is not the case. Instead, increasing resistance was found, and this trend was significant. 

### 3.3. Electrical Characteristics of E-Yarn

The approximate circuit of E-yarn with input voltage $V_{S}$ and internal SMD LED resistance R_LED_ are illustrated in Figure 4. The resistances of stainless steel yarn, crimp bead connecter, and SMDs are shown in a series configuration electrical circuit model in Figure 4A. The electrical resistance of the E-yarn was a direct function of the resistance of stainless steel conductive yarn, resistance of the crimp bead connector, and the resistances of SMDs. However, the resistance of the SMD LED is not governed by Kirchhoff’s 2nd law. The average overall electrical resistance of the left and right conductive yarn was 4.03 Ω (from Figure 5). The average LED SMD resistance and the total resistance of the SMD LED embed E-yarn measured at various input voltages beginning at 2.35 V and the forward current of 0.02 A were computed based on Equations (2) and (3). These values are presented in Table 5. The overall solder connector resistance was 0.277 Ω. These indicated that the soldering process between the silver crimp beads and tips of SMD LED was performed well, and there was not any deterioration found on the solder connection after the voltage and current flow in the circuit due to heat deception by the LED SMD. 

The characteristic I–V plot of the SMD LED-embedded E-yarn exhibits an oblique line showing that a small forward voltage change may result in a large change in current, as shown in Figure 7. The resistance of the SMD LED-embedded E-yarn was obtained from the current–voltage characteristic graph. From the graph it can be observed that if the forward voltage was less than 2.35 V, there was almost no flow of current and the resistance of the E-yarn was very high (2.2 kOhm); therefore, the LED was unlit. Between 2.35 and 2.5 V, the LED started to conduct. With more than 2.5 V forward voltage, the resistance of the E-yarn was completely reduced and the LED began to shine. From physical observation, we obtained that as the input voltage and current increased, the brightness of the LED increased. After 3.4 V forward voltage, the luminescence of the LED SMD light was decreased and subsequently the LED SMD to became completely damaged. From the results we obtained a yarn and connector resistance of 4.32 Ohm. The yarn length used was 30 cm, so this corresponds to 4.043 Ohm. As a consequence, we can conclude that the connector resistance of the SMD yarn was ($R_{\mathrm{LC}}+R_{\mathrm{RC}}$) 0.277 Ohm. This proves the connection made the LED with the E-yarn was very good and independent of the current.

### 3.4. Power of SS Conductive Yarn E-yarn

In addition to electrical resistance characterization, the performances of the conductive thread and E-yarn were investigated in terms of power and electrical resistance change as current levels increased. The performances of the SS conductive threads and SMD LED-embedded E-yarn in terms of their mean maximum attainable power before failure are shown in Figure 8A. They began to emit photons, which are small packets of visible light, and heated up, as shown in Figure 8.

The powers of the SS-conductive thread and E-yarn increased when the input voltage source increased from 2.35 to 3.4 V. The SMD LED-embedded E-yarn was able to sustain a power of 597.03 mW. These powers occurred due to the heat dissipation and light emission at the SMD LED electronic component. However, the power contribution of SS conductive yarn was negligible, as shown in Figure 8. The heat dissipation was recorded with an IR thermographic camera from Figure 8B, showing that only the LED increased in temperature after 10 sec of operation due to the SMD LED start to lit. However, the temperature profile of the SS conductive yarn (i.e spot 1 and spot 3) was near to room temperature. These indicated that, the heat generated by the SS conductive yarn relative to SMD component was insignificant and then the temperature was constant. According to these results, these manufactured E-yarns can be used for the development of wearable electronic textiles.

### 3.5. Effects of Washing on Electrical Conductivity 

The heat shrink tube was inserted onto the solder pad and was used to encapsulate the junction of the crimp beads and terminals of the SMDs. This was mainly to protect against mechanical abrasions on the solder pad, which was next subjected to 25 wash and dry cycles. The change in electrical resistance (CR%) was measured in all conductive stainless steel yarns, the standard SMD embedded E-yarn, and the E-yarn with encapsulated SMD junctions, after each periodical laundering cycle, as shown in Figure 9. The measuring of resistance of the E-yarn was performed with 3.0 V source voltage. The electrical resistance of the SS conductive yarns was much lower than that of the SMD LED-embedded E-yarn, as only the SS conductive yarn was present, which has low resistance. The change in the electrical resistance of the SS conductive yarn was not problematic, whereas the change in resistance of the E-yarn was. After 25 cycles of washing the CR% of the SS conductive yarn was 9.25%. As the initial resistance was low (see Figure 5), this had only a small influence on the efficiency of a circuit made with SS conductive yarn.

Additionally, Figure 9 shows that the E-yarns without heat shrink had a rapid increase in electrical resistance after 10 washes. All the samples were considered to have failed after 20 wash cycles. There was a complete lack of continuity due to partial fractures and breakages that occurred at or near the junctions between the connecting parts of the SMDs and the crimp beads. This implies that the solder connection point structures were strongly influenced by the mechanical action of the laundry [50]. Adding the heat shrink protection improved the situation for the E-yarn. Breakage no longer occurred, and at 25 cycles, the increase in resistance was 23.9%. In absolute terms, this is much higher than the SS yarn degradation and problematic for the correct working of the E-yarn. The increase occurred due to deterioration at the solder joints, leading to a forward voltage drop at the LED, and the resistance of the E-yarn increased. As a consequence, for the same source voltage, after repeated washing, the brightness of the LED dropped.

In conclusion, once SMD LEDs are integrated, it is possible to wash the samples 10 times, with or without heat shrink protection. After this, the degradation at the joints becomes too great, and the E-yarn should no longer be used. 

### 3.6. Tensile Strength of SS Conductive Yarn and E-Yarn

The specific load extension curves obtained for SS conductive yarn and SMD LED-embedded E-yarn samples are shown in Figure 10. Both the SS conductive yarn and E-yarn followed Hooke’s law. Tensile tests were performed on five samples of conductive SS yarn and five completed E-yarns containing one SMD with or without a heat shrink tube. 

Figure 10 shows that the conductive yarn and E-yarn acquired neither yield points nor strain hardening behavior (i.e., they followed a sharp plastic curve till they broke). This means that the ultimate tensile and breaking strengths of the used material were the same. Due to the straightening of the SS conductive thread structure at the necking point, the specific load extension curve steeply declined to a lower level before breakage. In Figure 10, a line graph is given before the breaking strength part. The SS conductive yarn, the E-yarn, and the heat shrink-encapsulated E-yarn were extended up to 3.9%, 2.9% and 3.9% before break, respectively. The average breakage forces were found to be 65.66, 54.44, and 59.09 N, correspondingly. Such stress-extension curves are typical for high performance fibers; they show high strength and low extensibility. The biggest difference is in tensile strength can be observed in the graph. We obtained that the conductive yarn had a tensile modulus of 75.57 cN/dtex, and the E-yarn without a heat shrink tube had a maximum tensile modulus of 66.07 cN/dtex. The heat shrink-encapsulated E-yarn had a maximum tensile modulus of 66.02 cN/dtex. The average values of strength characteristics of both stainless steel conductive yarn and LED-embedded E-yarn with and without a heat shrink tube are described in Table 6.

Based on the results shown so far, further investigations were performed by using an optical microscope, as shown in Figure 11. It is clear from Figure 11 that, during the tensile strength test, the failures of the SMD-embedded E-yarns were due to the breakage at the connection points between crimp beads and SMD terminals. The elongation at break of the E-yarns was less than that of the conductive yarn due to the formation of weak points in the solder joints between the terminals of SMDs and crimp beads. These weak points occurred due to the partial fracturing of the terminal parts of SMDs. However, the fractures’ effect on the pullout strength was considered to be very small. These weak points cannot hold the tensile stress relative to SS conductive yarns. Therefore, the strengths of the conductive SS yarn and the SMD E-yarn were significantly affected, though the resulting strengths were still sufficiently high for use in most e-textile applications. The heat-shrink-tube yarns gave better results that the pure E-yarn, but it should be noted that adding heat shrink tubes reduces the bending properties of the yarn. As seen in the previous section, this helps with washing resistance, but leads to a worse texture of the resulting yarn.

### 3.7. Proof of Concept

The brightness of the SMD’s display is essential to determining appropriate applications for a wearable textile for the reason that the luminosity of the display is dependent on this performance characteristic. In very bright situations, low-light displays are difficult to distinguish. Figure 12A shows an actual image of the fabricated SMD-embedded E-yarn. Three SMD-integrated stainless steel E-yarns were lit by supplying 7.68V DC input voltage with 0.05A current, verifying that the E-yarn was functional, as shown in Figure 12 B. The LEDs within the E-yarns provided great illumination. In addition, the fabricated E-yarn showed constant light emission despite bending stress on the conductive yarn, with any bending radius. This result shows that when compared to that of Kim et al. [46], the device tolerated greater bending stress. This proof of concept and all the above results regarding electrical conductivity and other E-yarn properties showed that the resulting E-yarn is promising for flexible, wearable textile sensors and actuators in items that do not require frequent washing.

## 4. Conclusions

In this research work, the integration of light-emitting surface mounted devices into stainless steel conductive thread with a combination of 1.5 mm crimp beads was performed. The techniques for insertion of SMDs into SS conductive yarn involved using hot air soldering method without damaging the SS conductive thread and the tiny SMDs. The investigation of electro-mechanical characteristics of the selected conductive yarns was performed. The influences of clamping gauge length and the strain of the conductive yarn on the electrical properties were studied for both the conductive yarn and SMD-integrated E-yarns. It is clear that as the clamping length increases, the resistance of the conductive yarn increases. The dependence of electrical resistance on clamping length showed that the SS conductive yarn has fairly linear behavior. According to the experimental data, the SS conductive yarn was built in a uniform manner resulting in no irregularities structurally. Furthermore, we presented data and findings on the effects of tensile strain on the electrical resistance of SS conductive threads and SMD-embedded E-yarn. The analytical finding showed that, due to the elastic deformation of the SS conductive yarn under strain, its cross-section decreased and the electrical resistance grew proportionally. In addition, the experimental results showed that breakage of all samples of E-yarn occurred at the connection point between the SMD terminal and crimp beads. This occurred likely due to poor mechanical bond formation during soldering process. All SMD-integrated E-yarns remained functional after twenty cycles of machine washing. The E-yarn, on the other hand, had a substantially greater failure rate before 20 washing cycles if its solder pad was not enclosed by a heat shrink tube. E-yarns’ ability to be washed is critical to their future use in the e-textile sector for wearable applications. This method enhanced the capacity to create E-yarns needed for the development of prototype electronic textiles, but washing resistance should be improved. One option might be using special washing techniques, but this would make electronic textiles less consumer friendly.

## Figures and Tables

**Figure 1 materials-15-02892-f001:**
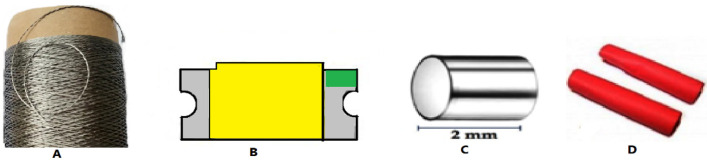
Image of stainless steel conductive yarn (**A**), SMD LED components (**B**), silver crimp beads (**C**), and heat shrink tube (**D**).

**Figure 2 materials-15-02892-f002:**
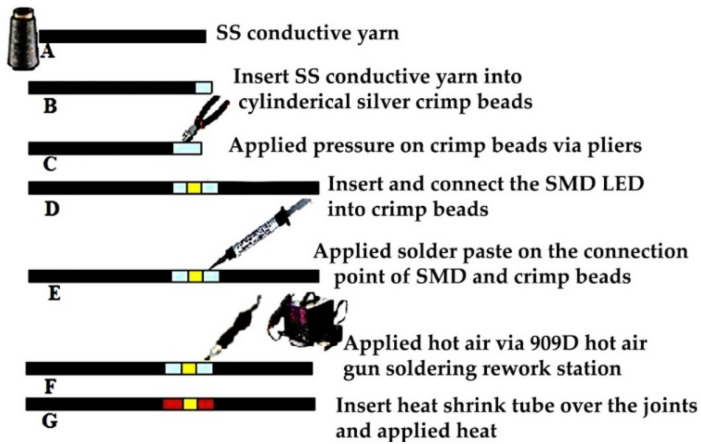
Steps to perform soldering.

**Figure 3 materials-15-02892-f003:**
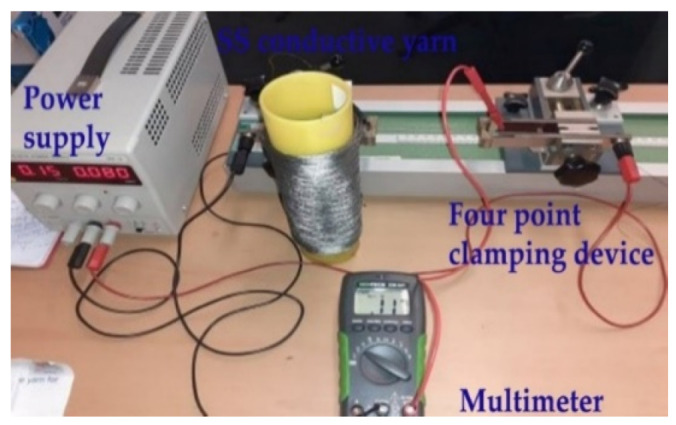
Measuring of length-dependent resistance (four-point clamping device).

**Figure 4 materials-15-02892-f004:**
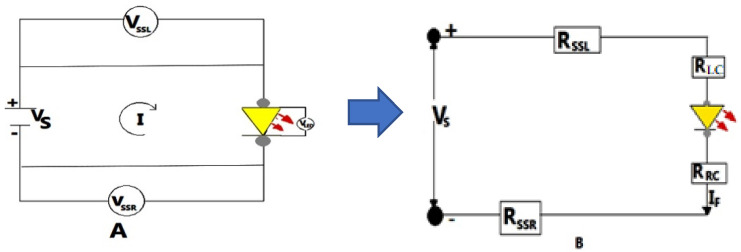
Schematic diagram of the LED connecting into SS conductive yarn (**A**) and total resistance measurement setup (**B**).

**Figure 5 materials-15-02892-f005:**
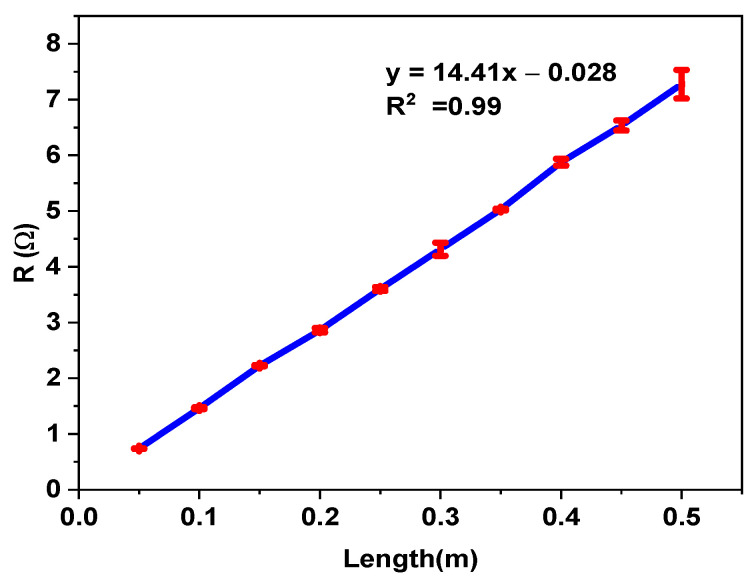
Length-dependent resistance of stainless steel conductive yarn.

**Figure 6 materials-15-02892-f006:**
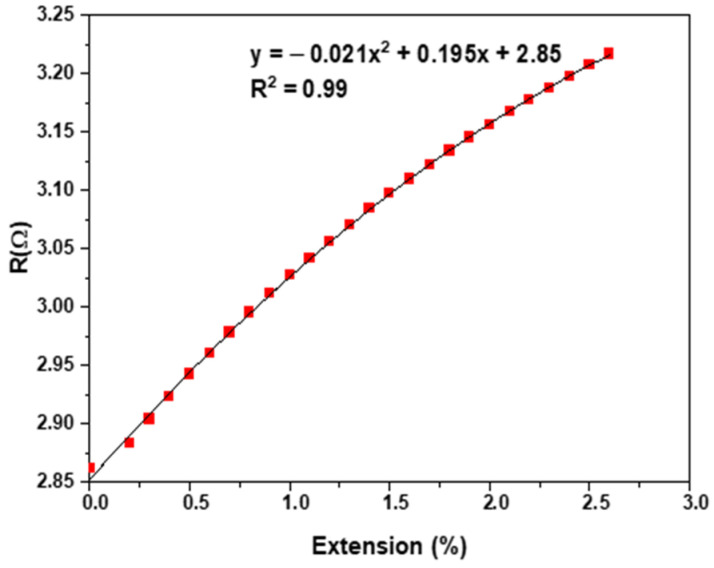
Stress-dependent electrical resistance of the conductive yarn.

**Figure 7 materials-15-02892-f007:**
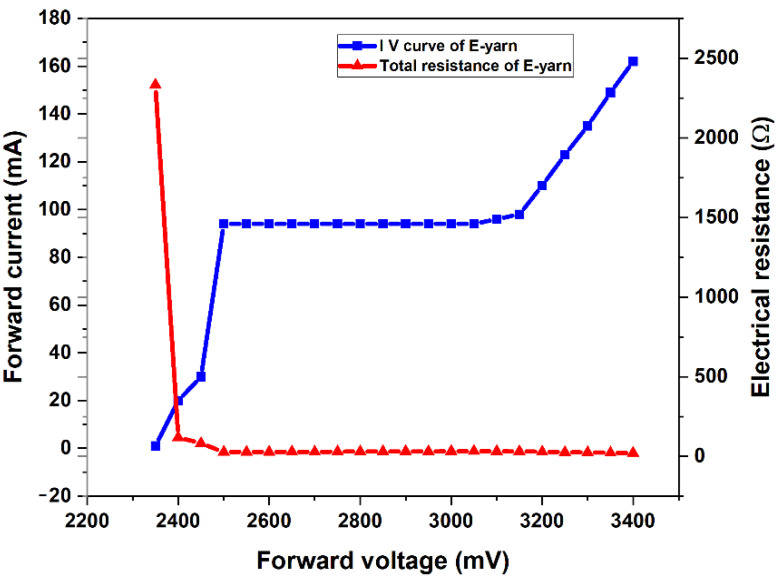
Current − voltage characteristic curve and total resistance of the E-yarn.

**Figure 8 materials-15-02892-f008:**
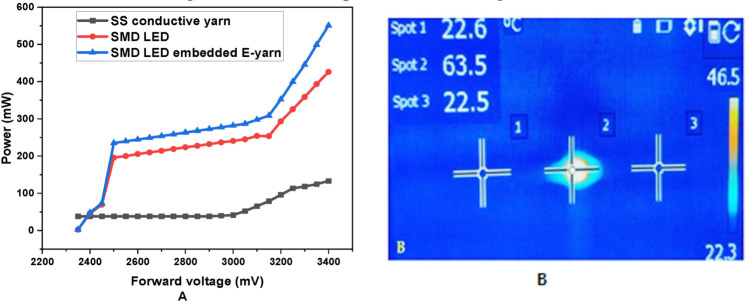
Power of SMD LED and E-yarn with a variable input voltage (**A**), IR thermographic image of conductive yarn (spot 1and spot 3) and brightness of LED SMD with 3.0 V (**B**).

**Figure 9 materials-15-02892-f009:**
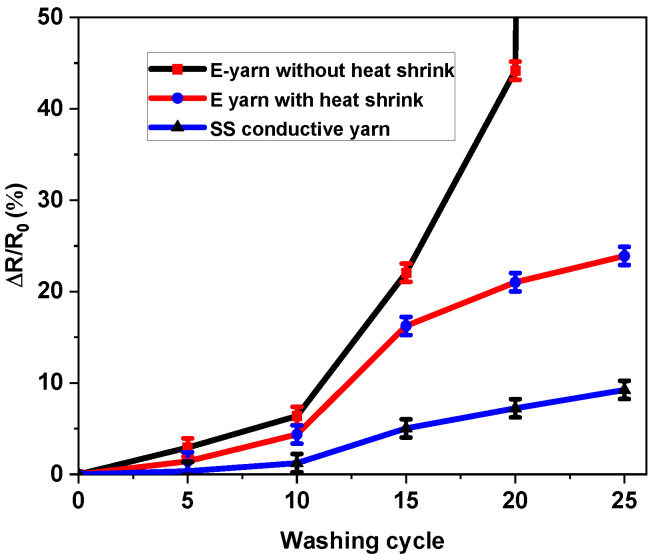
Effects of washing on the electrical resistance.

**Figure 10 materials-15-02892-f010:**
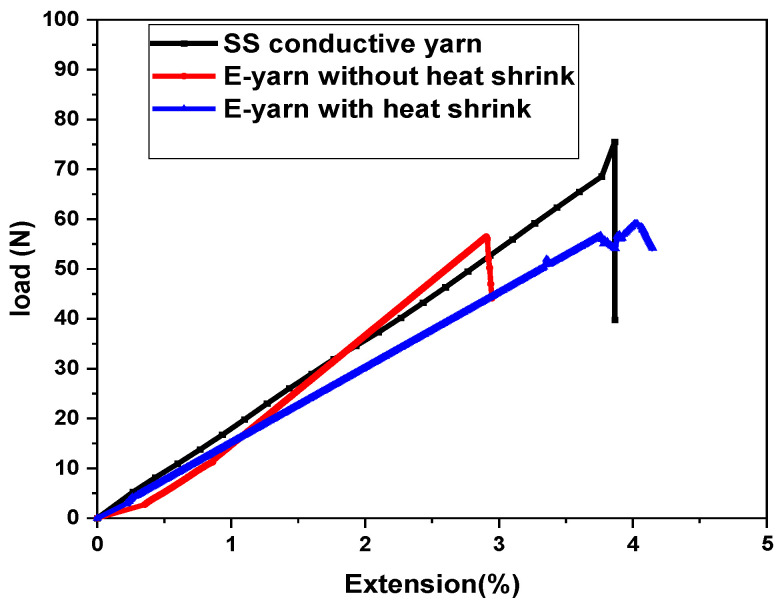
Strength of stainless steel conductive yarn and SMD LED-embedded E-yarns.

**Figure 11 materials-15-02892-f011:**
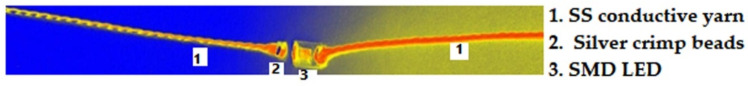
Breakage at connection point between the SMD LED and silver crimp beads.

**Figure 12 materials-15-02892-f012:**
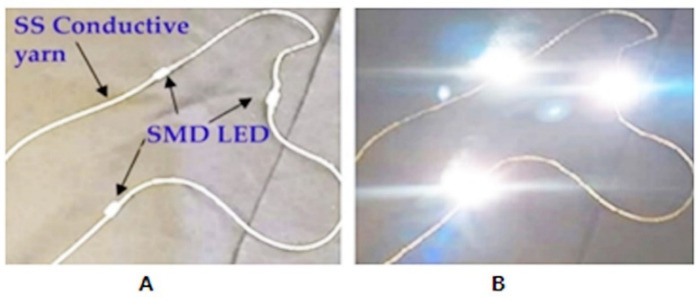
Actual prototype of LEDs integrated into a stainless steel yarn (**A**) and lit E-yarn (**B**).

**Table 1 materials-15-02892-t001:** ANOVA analysis of length vs. resistance of SS conductive yarn.

Source	SS	df	MS	No. obs. = 100
Model	433.47	1	435.076	Prob > F = 0.000
Residual	0.638	98	0.0065	R-squared = 0.99
Total	435.71	99	4.401	Root MSE = 0.081

**Table 2 materials-15-02892-t002:** Regression analysis of length vs. resistance of SS conductive yarn.

Resistance	Coef.	Std. Err.	t	*P* > t	[95% Conf. Interval]	
Length	14.41	0.056	258.47	0.000	14.41	14.64
_cons	−0.028	0.017	−0.01	0.989	−0.035	0.034

**Table 3 materials-15-02892-t003:** Regression analysis of extension vs. resistance of SS conductive yarn.

		Value	Standard Error	t-Value	Prob > |t|
B	Intercept	2.85	0.002	1660.46	5.5 × 10^-60^
	B1	0.195	0.003	66.26	9.13 × 10^-28^
	B2	−0.021	0.001	−19.76	6.29 × 10^-16^

**Table 4 materials-15-02892-t004:** ANOVA analysis of extension vs. resistance of SS conductive yarn.

		DF	Sum of Squares	Mean Square	F Value	Prob > F
B	Model	2	0.28823	0.14	17852.54	0.000
	Error	23	1.868 × 10^−4^	8.07 × 10^−6^		
	Total	25	0.29			

**Table 5 materials-15-02892-t005:** Total electrical resistance of a LED circuit with specific forward voltages (2.35–3.4V).

Source Voltage $\left(V_{S}\right)$ (mV)	LED SMD			E-yarn	
	Forward Current (I) (mA)	Forward LED Voltage (mV)	LED SMD Resistance (Ω)	Forward E-Yarn Voltage (mV)	Total E-Yarn Resistance (Ω)
2350	1	2246.60	2246.60	2333	2333.00
2400	20	2296.60	114.83	2383	119.15
2450	30	2312.40	77.08	2442	81.40
2500	94	2080.92	22.14	2487	26.46
2550	94	2127.92	22.64	2534	26.96
2600	94	2184.92	23.25	2591	27.56
2650	94	2231.92	23.75	2638	28.06
2700	94	2276.92	24.23	2683	28.54
2750	94	2327.92	24.77	2734	29.09
2800	94	2378.92	25.31	2785	29.63
2850	94	2420.92	25.76	2827	30.07
2900	94	2469.92	26.28	2876	30.60
2950	94	2518.92	26.80	2925	31.12
3000	94	2558.92	27.23	2965	31.54
3050	94	2605.92	27.73	3012	32.04
3100	96	2645.28	27.56	3060	31.88
3150	98	2586.34	26.39	3009.7	30.71
3200	110	2665.80	24.24	3141	28.55
3250	123	2648.64	21.54	3180	25.85
3300	135	2655.80	19.68	3239	23.99
3350	149	2639.32	17.72	3283	22.03
3400	162	2629.16	16.23	3329	20.55

**Table 6 materials-15-02892-t006:** Strength characteristics of SS conductive yarn and E-yarn.

Material	Modulus (cN/dtex)	Maximum Load (N)	Tenacity at Max Load (cN/tex)	Tensile strain at Max Load (%)
SS conductive yarn	75.57	65.66	11.9	3.8
E-yarn without heat shrink tube	66.07	56.44	9.8	2.9
E-yarn with heat shrink tube	66.02	59.09	10.1	3.9

## Data Availability

The data presented in this study are available on request from the corresponding author.

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
