# Peer review of "Development of Stainless Steel Yarn with Embedded Surface Mounted Light Emitting Diodes"

_materials, 2022, doi:10.3390/ma15082892_

Round 1
Reviewer 1 Report
In this research work, the integration of light emitting surface mounted devices into stainless steel conductive thread with a combination of 1.5 mm crimp beads was developed.
The investigation of electro-mechanical characteristics of the selected conductive yarns was performed. The influence of clamping gauge length and strain of conductive yarn on the electrical property was studied for both the conductive yarn and SMD components integrated E-yarn.
Furthermore, this study presents data and findings on the effect of tensile strain on the
electrical resistance of SS conductive threads and SMD component embedded in E-yarn.
Furthermore, all developed SMD components integrated E-yarn remained functional after twenty cycles machine wash. The E-yarn, on the other hand, had a substantially greater failure rate before 20 washing cycles if its solder pad was not enclosed by heat shrink tube. E-yarns' ability to be washed is critical to their future use in the E-textile sector for wearable applications. This method enhanced the capacity to create
E-yarn needed for the development of prototype electronic textiles, but mainly washing resistance should be improved.
The work is interesting and corresponds to the subject of a journal "Materials". However, before the article is accepted for publication, it is necessary to correct some shortcomings in the text.
1) Figures 4 a and b need to be improved in quality and the designations a and b should be added.
2) Figure 7A apparently stands for figure 7 on line 398.
3) In figures 8 and 12, it is necessary to enter the designations a and b.
In addition, there are vague sentences in the text. Authors need to carefully proofread the text and finalize the article.
Author Response
First of all, we appreciate your valuable comments given, and thank you for taking your precious time for the comments again. Your previous comments help us to improve our article. Thank you for your recommendation to publish our article in MDPI Material.

Reviewer 2 Report
This manuscript embedded SMD LEDs in stainless steel yarn to produce E-yarn by soldering and also characterized mechanical and washing cycles properties. A few points should be addressed before published.
- The quality of figures was poor and needs to be improved. The e-yarn and the soldering points should has clear images to show the details.
- In “LEDs (light emitting diodes) with dimension 1.6 × 1.2 × 20 mm (L × W × D)”, is it 2mm or 20mm? Wrong spelling of “siber” in “Figure 1. Image of stainless steel conductive yarn (A), SMD LED components (B), silber crimp beads “
- Too many citation of the websites, replace of scientific literatures
- Why silver crimp beads were needed in the research? How about the mechanical and electrical properties of the e-yarn without silver crimp beads.
Author Response

(The authors gave the same response as above.)
